# Proton-Pump Inhibitors in Eosinophilic Esophagitis: A Review Focused on the Role of Pharmacogenetics

**DOI:** 10.3390/pharmaceutics16040487

**Published:** 2024-04-02

**Authors:** Leticia Rodríguez-Alcolado, Pilar Navarro, Laura Arias-González, Elena Grueso-Navarro, Alfredo J. Lucendo, Emilio J. Laserna-Mendieta

**Affiliations:** 1Department of Gastroenterology, Hospital General de Tomelloso, 13700 Tomelloso, Spain; leticia3r@gmail.com (L.R.-A.); mpilar_ns@hotmail.com (P.N.); laura.arias.gonzalez@gmail.com (L.A.-G.); elenagru4@gmail.com (E.G.-N.); 2Department of Surgery, Medical and Social Sciences, Universidad de Alcalá, 28805 Alcalá de Henares, Spain; 3Instituto de Investigación Sanitaria de Castilla-La Mancha (IDISCAM), 45071 Toledo, Spain; 4Instituto de Investigación Sanitaria La Princesa, 28006 Madrid, Spain; 5Centro de Investigación Biomédica en Red de Enfermedades Hepáticas y Digestivas (CIBERehd), 28029 Madrid, Spain

**Keywords:** proton-pump inhibitors, eosinophilic esophagitis, pharmacogenetics, *CYP219* gene, *STAT6* gene

## Abstract

Proton-pump inhibitors (PPIs) are the most administered first-line treatment for eosinophilic esophagitis (EoE). However, only around half of EoE patients respond histologically to a double dosage of PPI. In addition, 70% of responders maintain EoE in remission after tapering the PPI dose. In order to avoid endoscopy with biopsies—the only accurate method of assessing PPI response—efforts have been made to identify PPI responder patients. The clinical or endoscopic features and biomarkers evaluated so far, however, have not proven to be sufficient in predicting PPI response. Although new approaches based on omics technologies have uncovered promising biomarkers, the specialized and complex procedures required are difficult to implement in clinical settings. Alternatively, PPI pharmacogenetics based on identifying variations in *CYP2C19* and *STAT6* genes have shown promising results in EoE, and could easily be performed in most laboratories. Other genetic variations have also been associated with PPI response and may explain those cases not related to *CYP2C19* or *STAT6*. Here, we provide an overview of PPI treatment in EoE and evidence of how genetic variations in *CYP2C19* and other genes could affect PPI effectiveness, and also discuss studies evaluating the role of pharmacogenetics in predicting PPI response in patients with EoE.

## 1. General Characteristics of Proton-Pump Inhibitors

Proton-pump inhibitors (PPIs) are a class of acid-suppressing agents, which are among the most utilized drugs worldwide. They are used to treat gastroesophageal disorders, predominantly gastroesophageal reflux disease (GERD), peptic ulcer disease, Helicobacter pylori infection, erosive esophagitis, Zollinger–Ellison syndrome, and eosinophilic esophagitis (EoE) [1].

PPIs are benzimidazole derivates which, after being absorbed in the small intestine, become its active form in the gastric parietal cells; they irreversibly block the gastric H+/K+ adenosine triphosphatase (ATPase) pump after covalently binding to cysteine residues, thus inhibiting acid secretion. This inhibition is only reversible through the production of new proton pumps, which can take up to 54 h [2]. Despite their short half-life of approximately one hour, the effect of PPIs lasts for 48 h and reaches an acid suppression steady-state in 2–3 days, meaning they are capable of inhibiting around 70% of daily acid production in the stomach [2].

The main metabolism pathway of PPIs is cytochrome P450 (CYP450), specifically CYP2C19 and, to a lesser extent, CYP3A4/5 enzymes [3]. These two enzymes mediate the hydroxylation and sulfoxidation of PPIs required for their clearance, and determine the pharmacokinetic (PK) and pharmacodynamic (PD) profile of PPIs [4].

There are five PPI drugs approved in most developed countries: the first-generation PPIs include omeprazole, lansoprazole, and pantoprazole, while the second-generation consists of esomeprazole (a stereoisomer of omeprazole) and rabeprazole. There are also two further second-generation PPIs approved in some countries: dexlansoprazole in the USA and ilaprazole in Korea and China. Second-generation PPIs are more effective, faster in achieving acid suppression, and less dependent on the CYP450 enzymatic metabolism [5].

Although they are proven to be safe and their prescription has been extended to other diseases, some side effects have been noted regarding their long-term use; these include an increased risk of osteoporosis in post-menopausal women, kidney damage, increased risk of certain infections (pneumonia and Clostridium difficile), and nutritional deficiencies or lower counts of platelets and hemoglobin [6]. However, most of the putative adverse outcomes associated with PPI use are not supported by high-quality evidence and are likely to have been affected by underlying confounding factors [7,8].

## 2. PPI Treatment in Eosinophilic Esophagitis

### 2.1. Brief Description of EoE Pathophysiology

EoE is a chronic, local immunity-mediated esophageal disorder, characterized clinically by symptoms of esophageal dysfunction, and histologically by an eosinophil-predominant inflammation restricted to the esophagus [9], defined by ≥15 eosinophils per high-power field (eos/HPF) at any esophageal level. First described three decades ago [10], EoE currently represents the leading cause of dysphagia and food impaction among children and young adults [11]. Patients with EoE commonly have concomitant atopies, resulting in allergy being involved in the origin of the disease [12]; indeed, EoE was initially characterized as a particular form of non-IgE-mediated food allergy [13,14]. An increased expression of T-helper (Th)-2 cytokines in the esophageal inflammatory infiltrate, including interleukin (IL)-5, IL-4, IL-13 and thymic stromal lymphopoietin (TSLP), is involved in the pathophysiology of EoE [15], as in other type 2 inflammatory diseases [12]. These cytokines are responsible for lymphocytes’ differentiation, Th2 polarization in the esophageal mucosa (IL-4), the proliferation, maturation and release of eosinophils from bone marrow (IL-5), the production and release of eotaxins, which are potent eosinophil chemoattractants (IL-13), the increased permeability of the epithelial barrier (IL-3), and the maturation of antigen presenting cells (TSLP), among other functions [16]. Eotaxin-3 is the most upregulated gene in the esophageal mucosa of EoE patients [17], with its transcription depending on the STAT6 nuclear factor [18]. This results in a long-lasting inflammatory response, which affects the different layers of the esophageal wall [19], causing esophageal dysmotility [20] and promoting a fibrous remodeling that may progress into esophageal strictures [21,22].

Apart from dietary therapy to avoid food culprits from triggering and maintaining esophageal inflammation, drug therapy, mainly based on topical corticosteroids and PPIs [23], has been used to treat EoE since the first descriptions of the disease. Due to their wide availability, easy administration, convenience, and positive safety profile, PPIs represent the most commonly used first-line therapy for EoE—as repeatedly documented by a series of patients of all ages and from different settings [24,25,26,27,28,29].

### 2.2. The Evolving Concept of PPI Response in EoE

The presence of eosinophils in the esophageal mucosa was erroneously linked to GERD in the early literature [30,31]; while no other effect of PPIs was known beyond their suppression of gastric secretion, the disappearance of the eosinophilic infiltrate in esophageal biopsies after PPI treatment suggested that GERD was its cause. A lack of response, or alternatively a normal esophageal pH monitoring, were required to diagnose EoE [32]. However, a prospective series in 2011 revealed that the clinical, endoscopic, and histological features of EoE were not distinguishable between patients who did and did not respond to PPI therapy; whereas in a subset of patients who resolved eosinophilic infiltration in esophageal biopsies, GERD could not be demonstrated by esophageal pH-monitoring, thus dissociating the response to PPI from GERD-associated acid exposure [33]. This gave rise to a provisional entity called “PPI-responsive esophageal eosinophilia”, or PPI-REE, to define those patients who resolved an apparent food allergy through the use of a drug that is able to block gastric acid secretion [34].

Over the next few years, cumulative evidence demonstrated that patients with so-called PPI-REE and those with “classic” EoE were identical at baseline in terms of symptoms [35,36], endoscopy appearance [35,36,37], and esophageal biopsy features [36,38,39], and even showed the same altered gene expression in esophageal samples [40]. Furthermore, among responders, PPI therapy downregulated Th2 cytokines’ gene expression [41], and reversed the abnormal EoE gene transcriptomic signature [40] and the changes induced by IL-13 responses [42], in the same way as when EoE patients swallowed topical steroids. Consequently, an international position paper [43], and European [9] and American [44] guidelines recognized PPI therapy as a true first-line therapy for patients with EoE.

### 2.3. PPI Dosages, Treatment Duration, and Effectiveness in Inducing Remission

By analogy with reflux disease, double doses of PPI for a period of 8 weeks were initially proposed for the treatment of patients with EoE [34]. This therapy proved effective in curing peptic erosions in most patients with erosive GERD [45] and was also considered suitable for EoE.

The first prospective study that systematically evaluated the response to double-dose PPI therapy in these patients revealed a 50% histological response—defined by <15 eos/HPF [33]. This rate of effectiveness was reproduced in two small randomized controlled trials [46,47] and by the first meta-analysis of 33 studies with 619, mostly European and US, EoE patients [48]. Large registries, based on data obtained from real-world practice, have been recently made available: a retrospective study in 236 adult patients from Denmark disclosed histological remission in 49% of patients after treatment with an 8-week high-dose PPI trial [49] and prospective data from the EoE CONNECT registry on 630 European patients reproduced an overall clinical plus histological remission rate of 49% [24], with both studies defining histological remission as a peak eosinophil count below 15 eos/HPF. When higher histological criteria (<5 eos/HPF) were considered, remission was achieved by 33–40% of patients after double PPI doses [24,46,47]. In addition, in those PPI-responsive patients, PPIs proved effective in reversing the endoscopic features of fibrosis and in improving esophageal distensibility [50].

The effectiveness of PPI in pediatric EoE patients has been shown to be similar, although the studies have, in general, presented more heterogeneous results, most likely due to their smaller sample sizes. In the first prospective study conducted in 51 children treated with high-dose PPI (esomeprazole 1 mg/kg, twice daily), 68% were found to have <15 eos/HPF after an 8-week trial [51]. The abovementioned meta-analysis included 188 children, among whom PPI therapy resulted in a summary estimate effectiveness of 54% [48]. More recently, an analysis of the prospective Spanish nationwide RENESE registry found that histological and clinico-histological remission was observed in 51.4% and 46.5% of the 346 children included [52].

To optimize PPI-effectiveness, extending the treatment duration up to 12 weeks has been shown to increase effectiveness by up to 65.2% (odds ratio = 2.7, 95% CI: 1.3–5.3, compared to treatment between 8 and 10 weeks) [24]. The use of double doses in induction, compared to the standard, determined the effectiveness in children [50] and adults [23]. In contrast, no significant differences in remission rates were shown for the different PPI drugs when used at equivalent doses [24]. Dividing the total PPI dose into two intakes also showed a non-significant trend in increasing its effectiveness in remission compared to once-daily dosing [48].

### 2.4. Long-Term Maintenance Therapy with EoE: Effectiveness and Monitoring

Several observational studies have provided consistent data on the effectiveness of tapering doses of PPIs in maintaining long-term EoE remission among initially responding patients. In adults, 73–81% of patients remained in remission after 1 year with half the effective induction dose [53,54]. For pediatric patients, 70–78% of initial responders to double doses remained in clinical and histological remission on tapering maintenance doses of 1 mg/kg/day after one year [51,55]. These figures were reproduced by large registries of clinical practice [24,52].

Importantly, relapsing esophageal inflammation (>15 eos/HPF) on tapering PPI doses among initial PPI responders can be effectively managed by resuming the initial higher doses [53]. Therefore, only a minimal proportion of initial PPI responders may require high-dose maintenance PPI to ensure a sustained response.

In terms of histological remission [56], EoE symptoms have repeatedly been shown not to be reliable enough to be used to monitor treatment response; for PPI therapy, clinical remission rates repeatedly exceed those of histological remission. Despite considerable effort to identify non- or minimally invasive methods to assess esophageal inflammation in EoE [57], endoscopy with biopsies currently remains the only accurate procedure. Therefore, a follow-up assessment based on endoscopy with biopsies should be performed 8–12 weeks after the initiation of any induction treatment based on PPIs for active EoE, or after any major treatment change (e.g., dose reduction or withdrawal of maintenance therapy) [58].

### 2.5. Clinical Predictors of PPI Effectiveness in EoE

The initial response to PPI therapy has been associated with some clinical and endoscopic aspects. Neither the age nor the sex of the patient determines the effectiveness of this therapy, nor does its use as the first or as a subsequent line of treatment (after failure of dietary therapy or swallowed topical corticosteroids) [52,59,60]. On the other hand, an increased body mass index was found to reduce the chances of achieving remission with PPIs [61]. The presence of fibro-stricturing endoscopic features (esophageal rings or strictures) has consistently been shown to reduce the chances of response to PPI [24,52,59,62], which could explain why patients who debuted with food impaction also had a higher probability of PPI therapy failure (adjusted odds ratio = 2.8, 95% confidence interval [CI]: 1.1–7.4) [63]. Edema and vertical lines are other endoscopic features that are found more commonly in children who did not respond to PPIs [64].

Regarding histopathological findings, a higher score in the EoE Histology Scoring System in the middle esophagus [65] and a lower immunostaining score of filaggrin [66] were associated with reduced PPI response.

The frequency or type of concomitant atopy to EoE has not been shown to determine response to PPI [52,62]; however, higher peripheral eosinophilia at baseline independently predicted failure to PPI response in adult EoE patients in two retrospective series [61,63], and rhinoconjunctivitis (odds ratio = 8.6, 95% CI: 1.5–48.7) was found to be a predictor of loss of response to PPIs during maintenance after dose reduction [53]. In addition, higher levels of eosinophil-derived neurotoxin (EDN) were found in samples from the esophageal brushing of child/young adult PPI non-responders before treatment compared to responders [67].

In patients who initially responded to PPI, some studies have suggested that sustained remission on tapering PPI doses was more common in those patients who achieved a deep remission (<5 eos/HPF), as compared to those with partial remission (5–14 eos/HPF) [24,55,68].

### 2.6. Prediction of PPI Response by Omics Studies

The wide access to omics technologies has favored the application of these methodologies in studies trying to determine which patients will respond to PPIs at the point of EoE diagnosis. The first studies evaluated the transcriptome profile, but no differences were found [40,69]; a result that was later confirmed [70,71].

Although transcriptomics was unsuccessful in classifying patients into PPI responders or non-responders, a gene expression analysis did identify that EoE patients could be classified into three endotypes according to the results of the EoE Diagnostic Panel, composed of 95 genes [72]. Using this strategy, it might be expected that patients with the milder EoE endotype (named EoEe1) would have a greater probability of responding to PPIs than the other two endotypes.

Other approaches have also been successful in this purpose. For example, a study in a population of 39 children with EoE identified a profile of esophageal micro-RNAs, composed of miR-7-5p, miR-375-3p, and miR-223-3p, that was able to predict the response to esomeprazole with an area under the ROC curve (AUC) of 0.90 [73].

Similarly, our group collaborated in a study that discovered a proteomic signature composed of 28 proteins that were differentially accumulated in esophageal biopsies from responder and non-responder patients [71]. This study also confirmed previous evidence with regard to there being no difference between PPI responders and non-responders in transcriptomic analyses.

The main limitation for the clinical implementation of these three strategies is that they still require endoscopy, since esophageal biopsy is the starting material, and the specialized techniques needed for gene/miRNA/protein characterization.

## 3. Role of CYP2C19 in Response to PPI

Cytochrome CYP2C19 is an enzyme, and part of the CYP450 super-family that contributes to the metabolism of many drugs, including antidepressants, benzodiazepines, mephenytoin, clopidogrel, and, as previously mentioned, PPIs [74]. Variations in the gene encoding for CYP2C19 are the most important and well-studied pharmacogenetic factors affecting response to PPIs. Despite other possible factors, the *CYP2C19* genotype is responsible for a large percentage of the PK variability regarding PPIs [75].

### 3.1. Genetic Variants, Phenotypes, and Populations’ Frequencies

The *CYP2C19* gene is located on chromosome 10p23.33 and is highly polymorphic, with 36 known variant haplotypes (star (*) alleles) and a plethora of diplotypes with a continuous level of activity [76,77]. These alleles are classified into functional groups, which are based on in vivo or in vitro information, when available, as follows: normal function (e.g., *CYP2C19*1*); no function (e.g., *CYP2C19*2*, **3*, **4*); decreased function (e.g., *CYP2C19*9*); increased function (*CYP2C19*17*); and uncertain function (e.g., *CYP2C19*12*, **14*). Table 1 summarizes the main genetic variations of *CYP2C19*.

The frequency of these alleles varies between populations [77]. Among the non-functional alleles, *CYP2C19*2* is the most common, with a frequency of approximately 60% in Oceanians, 27–29% in Asians, 12–18% in Europeans, Africans, and Americans, and around 11% in populations with Latino or Near Eastern origins. *CYP2C19*3* also has a high presence in some populations, with an allele frequency of 15% in Oceanians, and 2–7% in Asians, but is rarely found in other populations. Most of the other variant alleles have a frequency below 1%. Increased-function allele *CYP2C19*17* is present in 15–20% of African, European, Near Eastern, and Latino populations, but in only 2% of East Asians and 5% of Oceanians. According to data from 2.29 million people in a study assessing the three most frequent variants (*CYP2C19*2*, **3*, and **17*), 58.3% of the participants expressed at least one increased or decreased allele function [78]. However, these allele frequency data may not be entirely accurate, as they are based on published data that are limited for some populations and most studies only test for the most common alleles, which can lead to certain alleles being underestimated [76].

The combination of different alleles of the *CYP2C19* gene leads to various phenotypes that are categorized according to the enzyme’s activity level (Table 2) [3,77]. Individuals with two normal function alleles are classified as CYP2C19 normal metabolizers (NM), and those individuals carrying two non-function alleles are classified as CYP2C19 poor metabolizers (PM). Individuals with one normal and one non-function allele or one non-function and one increased-function allele are considered CYP2C19 intermediate metabolizers (IM). The data suggest that the loss of function caused by allele *2 has a much greater impact on the phenotype than the gain of function of allele *17, resulting in an IM phenotype [79]. Individuals carrying one normal and one increased-function allele are classified as CYP2C19 rapid metabolizers (RM) and those carrying two increased-function alleles are considered CYP2C19 ultrarapid metabolizers (UM). As limited data are available for decreased function alleles, individuals carrying one decreased-function and one non-function allele are classified as “likely PM” and those carrying one normal-function and one decreased-function allele, or one increased-function and one decreased-function allele, or two decreased-function alleles, are currently classified as “likely IM”. Finally, individuals carrying one or two uncertain-function alleles are assigned an “indetermined metabolizer” phenotype.

The frequency of these phenotypes also differs among populations [77]. Phenotypes with non-function alleles, such as CYP2C19 PM and IM, have the highest prevalence in East Asia (13% and 46%, respectively) and Oceania (57% and 37%, respectively), and are less common in Europe (2% for PM and 26% for IM) and Africa (4% for PM and 31% for IM). Conversely, RM and UM phenotypes have a high prevalence in Europeans and Near Eastern populations (25–27% and 3–4%, respectively), and are rarely found in other populations, such as East Asians (2% and 0.5%, respectively). Overall, most individuals in any part of the world have a phenotype other than NM (between 48 and 67%) [80].

### 3.2. Clinical Implications of CYP2C19 Phenotypes

The impact of the *CYP2C19* genotype on the PK and PD characteristics of PPIs varies depending on the contribution of this enzyme to the metabolism of each PPI drug [75]. In first-generation PPIs (omeprazole, lansoprazole, and pantoprazole), CYP2C19 is responsible for more than 80% of their metabolism [81], making these drugs more susceptible to the impact of *CYP2C19* genetic variations. The metabolism of the second-generation PPI esomeprazole is less reliant on CYP2C19 compared to omeprazole [82], and is therefore less affected by genetic variations in this enzyme. Rabeprazole, the other second-generation PPI, is the least influenced by the genetic variations in *CYP2C19*, as it is mainly metabolized via a non-enzymatic pathway [83].

Several studies have shown that individuals with CYP2C19 IM and PM phenotypes exhibit decreased clearance and increased PPI plasma concentrations when compared to NMs, which results in higher PPI exposure and leads to a more pronounced acid suppression effect, as measured by intragastric pH [3,75]. The AUC of omeprazole and lansoprazole was 4–12-fold higher in PM than in NM phenotype carriers, and in the case of pantoprazole and rabeprazole, the AUC was 6- and 2-fold higher in PM than in NM, respectively [75]. The median intra-gastric pH was also higher in PM compared to the other phenotypes when standard doses were given [84]. In contrast, those individuals with CYP2C19 RM and UM phenotypes have increased clearance and decreased plasma concentration compared to NM, resulting in lower PPI exposure, which may lead to an increased risk of treatment failure [3]. For these phenotypes, there are less data regarding the association with the PK/PD parameters of PPIs. This is due to the fact that most studies were carried out in populations with a low prevalence of the *CYP2C19*17* allele or conducted prior to its discovery [85]. However, it has been reported that those with RM and UM phenotypes have a lower AUC than those with NM, IM, and PM [3,75,84].

The resulting different levels of exposure to the drug influence PPI effectiveness in the treatment of several diseases. In the case of GERD, a meta-analysis that included 19 studies demonstrated that the efficacy rates of PPIs varied significantly among CYP2C19 phenotypes (52.2% in NMs; 56.7% in IMs; 61.3% in PMs; *p* = 0.047) and that those subjects carrying a RM phenotype had an increased risk of being refractory to PPI therapy when compared with PMs (odds ratio = 1.7, 95% CI: 1.0–2.7) [86].

For the efficacy of H. pylori infection eradication therapy, a meta-analysis including 39 studies showed that, regardless of the type of PPI used, the treatment duration, or the treatment regimens, there were significant differences in eradication rates according to CYP2C19 phenotypes between extensive metabolizers (EM, which includes those with NM or RM phenotypes) and IMs (79.2% vs. 84.0%, odds ratio = 0.7, 95 CI: 0.6–0.9,), and also between EMs and PMs (79.2% vs. 87.0%, odds ratio = 0.6, 95 CI: 0.5–0.7), but not between IMs and PMs [87]. Furthermore, when different PPI drugs were evaluated, these differences were found in patients treated with omeprazole, lansoprazole, and esomeprazole but not in those treated with rabeprazole (due to its mainly non-enzymatic metabolism) or pantoprazole (probably due to the low number of studies that evaluated this drug) [87].

As the influence of the CYP2C19 phenotype on the clinical efficacy of PPI treatment is well-documented, some clinical guidelines have included recommendations to adjust PPI dosage in certain diseases. The Clinical Pharmacogenetics Implementation Consortium (CPIC) established evidence-based guidelines, which included strong recommendations for most CYP2C19 phenotypes when treated with first-generation PPIs [3]. Also, guidelines from the Dutch Pharmacogenetics Working Group (DPWG) made recommendations according to patients’ CYP2C19 phenotype, but limited these recommendations to the usage of first-generation PPIs in H. pylori eradication, affecting only the UM phenotype [88].

## 4. Other Genetic Variations Influencing Response to PPIs

### 4.1. CYP2C18

The association between a haplotype of two SNPs close to the *CYP2C18* gene (rs2860840 C > T and rs11188059 G > A), *CYP2C:TG*, and a UM CYP2C19 phenotype was recently established by assessing escitalopram metabolism [89]. One year later, another group found that the same haplotype was also associated with treatment failure of omeprazole in GERD [90]. Specifically, they described a higher proportion of homozygous patients for *CYP2C:TG* compared to the reference population (New Zealand European) among patients with objective GERD, despite treatment with omeprazole (≥40 mg/day) for a minimum of 8 weeks (*p* = 0.03).

Interestingly, all homozygous *CYP2C:TG/TG* patients did not have the variant *CYP2C19*17* in both studies, thus suggesting that this haplotype could be a new genetic variant, which may explain the rapid metabolism of PPIs in patients lacking the *CYP2C19*17* allele.

### 4.2. CYP3A4/5

As mentioned previously, the CYP3A family represents a secondary enzyme participating in the metabolism of most PPIs. The two main enzymes are CYP3A4 and CYP3A5, which are also involved in the biotransformation of multiple drugs (anti-depressants, calcium antagonists, immunosuppressants, opiates, statins, steroids, etc.) [91]. Although genetic variations are rare for *CYP3A4*, a variant for *CYP3A5* implying a splicing defect is common in all populations, except the African population [92]. The most studied variations are *CYP3A4*22* (rs35599367 C > T), which causes decreased function [93], and *CYP3A5*3* (rs776746 T > C), which showed a lack of activity [94].

To date, no proven relevant effect of genetic variants in *CYP3A4/5* on PPI metabolism has been found, excepting the *CYP3A5*3/*3* genotype’s influence on ilaprazole clearance, described in a Chinese population [95]. However, *CYP3A4/5* variants could be important in certain situations due to drug–drug–gene interactions [80]. For example, the proportion of PPI metabolized by CYP3A is higher in CYP2C19 IMs and PMs [96,97]; thus, treatments with other drugs metabolized by CYP3A could mutually affect their concentrations, or concomitant treatment with any drug inhibiting CYP3A could lead to an increased risk of adverse PPI events [98]. Another case in which *CYP3A4/5* genetic variants could have more influence on PPI response is CYP2C19 inhibition by drugs such as fluvoxamine. As rabeprazole is rarely metabolized by these pathways, it is suggested that this PPI could be chosen for patients concomitantly treated with drugs metabolized by CYP2C19 and/or CYP3A, or drugs inhibiting these enzymes [98].

### 4.3. ABCB1

The *ABCB1* gene, (ATP-binding cassette, sub-family B, member 1; formerly known as multidrug-resistance transporter gene 1 or MDR1), codifies a P-glycoprotein involved in the absorption of PPIs in the small intestine. This protein is also involved in the bioavailability of multiple drugs, and some of its genetic variations have clinical implications [99].

One of its most studied SNPs, rs1045642 (C3435T), was investigated in two studies assessing its influence in lansoprazole PK and PD. In the first of these, higher plasma levels of lansoprazole were found in 15 Japanese subjects (all *CYP2C19*1/*1*) with the rs1045642-TT genotype, but no effect was observed regarding intragastric pH [100]. In the other study, including 24 healthy Chinese volunteers, a trend toward the improved absorption and rapid elimination of rs1045642 wild-type subjects was detected, while the effect on PK parameters was significant for different *CYP2C19* genotypes [101]. Therefore, according to these results, although rs1045642 could have a role in PPI effectiveness, its effect would be minor and less important than that exerted by *CYP2C19* variations.

### 4.4. ATP4A

The gastric H+/K+-ATPase pump is responsible for generating the acidic environment in the stomach and is the main target of PPIs [102]. Therefore, it makes sense that genetic variations of this pump could affect PPI effectiveness. This issue was investigated by a Chinese group, which analyzed the influence of the rs2733743 (*ATP4A* A > G) in acid suppression by dexlansoprazole injections in 51 healthy subjects [103].

Firstly, they found that this variation was quite common in their population, with 35% in heterozygosis and 38% in homozygosis. Secondly, they compared gastric acid inhibition among different genetic variations for *CYP2C19*, *ABCB1,* and the aforementioned rs2733743, and discovered that the inhibitory effect was affected by *CYP2C19* genotypes and rs2733743 homozygotes (GG genotype), with the latter showing a greater inhibition [103]. To date, no other studies have confirmed this association in other populations.

### 4.5. STAT6

Signal Transducer and Activator of Transcription 6 (STAT6) is a mediator of T helper type 2 cell response and its synthesis is stimulated by certain interleukins, mainly IL-4 and IL-13. STAT6 thus plays an important role in atopy and allergic diseases, including EoE [104,105]. Eosinophilic infiltration in the esophagus is led by eotaxin-3, whose expression is stimulated by STAT6. As PPIs can block STAT6 binding to the eotaxin-3 promoter [106], it is feasible that genetic variations in STAT6 could affect PPI response in EoE.

This hypothesis was investigated by Mougey et al. in two studies that recruited children with EoE, in whom eight SNPs of *STAT6* were determined [68,107]. Their results are described in detail in the next section of this review and summarized along with the other genetic variations affecting PPI response in Table 3.

## 5. Pharmacogenetic Studies on PPI Effectiveness in EoE

To date, only four studies—three full papers and one congress abstract—have evaluated the role of pharmacogenetics in PPI response in EoE. Two of these analyzed *CYP2C19* genotypes only, while the other two, performed by the same research group, determined genetic variants in *CYP2C19* and *STAT6*.

The first study analyzing the influence of pharmacogenetics in the response to PPIs in EoE was carried out in 2015 [53]. In this study, 75 adult patients with EoE, from eight hospitals in four different countries, were included. *CYP2C19* was genotyped in 50 patients who initially achieved remission with PPIs, and its association with loss of histological response after dose reduction was evaluated. Maintenance treatment length until effectiveness assessment was highly variable (minimum one year; mean of 26 months), as were the PPI drugs used (omeprazole, esomeprazole, and pantoprazole) and dosages (60% double doses and 40% single dose). In univariate analysis, subjects with *1/*1 genotype and those with at least one *17 allele (66% of patients) were more frequent in the group of patients who experience EoE recurrence (36% vs. 6%). In the multivariate model, these patients showed a 12.5-fold increase in the odds of losing response.

The next two studies were undertaken by the same researchers in a cohort of children with EoE from two Spanish hospitals and evaluated the influence of *CYP2C19* and *STAT6* in the histological response to PPIs [68,107].

In the first study, 92 children were included, mainly treated with esomeprazole (96%) at different doses (ranging between 0.46 and 2.40 mg/Kg/day) to induce EoE remission [107]. Regarding *CYP2C19*, subjects carrying the *CYP2C19*17* allele had a 7.7 times greater probability of not responding to PPIs (*p* = 0.031). Interesting findings were observed for STAT6 variations. Firstly, SNP rs324011 (which is in linkage disequilibrium with rs167769 and rs12368672) showed an association with the peak of eos/HPF in distal biopsies before treatment, with a 1.7-fold increase in subjects with TT genotype (*p* = 0.048). In addition, this SNP showed a synergistic effect with *CYP2C19*17*, containing this allele and one or two copies of the variant for rs324011 (genotypes CT/TT), which increased the odds of not responding to the PPIs by 8.7 times (*p* = 0.022). Another SNP, the rs1059513, also displayed relevant results, carrying one or two copies of the variant (genotypes TC/CC), and was independently associated with response to PPIs (*p* = 0.028), with a 6.2-fold increase in the achievement of histological response in the full cohort and 14.9-fold better odds for individuals who do not carry the *CYP2C19*17* allele.

In the study by Mougey et al., performed two years later [68], a group of 73 child responders to PPI (mostly from the same cohort as the previous study), who followed a dose reduction in their PPI treatment to maintain remission (dose range for maintenance was 0.23–1.22 mg/Kg/day), were included. An endoscopic assessment was carried out after 1 year and the influence of variants in *CYP2C19* and *STAT6* were again analyzed. The CT/TT genotypes of rs324011 showed a higher probability of EoE relapse, with a 2.8-fold increase in the chance of having ≥15 eos/HPF after PPI reduction (*p* = 0.029). A similar effect was observed for the other two SNPs in linkage disequilibrium (rs167769 and rs12368672; *p* = 0.060 and *p* = 0.021, respectively). Interestingly, they did not detect significant associations between *CYP2C19*17* (or combinations of this allele with the former three SNPs of STAT6) and maintenance of EoE remission.

Although they provided relevant data about the effect of *CYP2C19* and *STAT* variants in the response to PPIs, for both the induction and maintenance of remission in EoE, the studies of Mougey et al. had limitations. These included a small sample size, variations in PPI dose and the length of therapy for induction, the inclusion of only pediatric populations, and not assessing the adherence to PPI therapy.

Finally, Bortolin et al. described *CYP2C19* genotyping in a group of 37 Canadian children with EoE, and found a high proportion of RM and UM variants (32% and 11%, respectively) [108]. In this cohort, pharmacogenetic testing to guide PPI dosing resulted in a treatment change in 78% of patients—mainly a dose increase and PPI switch to rabeprazole in RM- (25% dose increase and 42% PPI switch) and UM- (100% PPI switch) carrier children. However, results relating to effectiveness before and after *CYP2C19* genotyping were not available, and have not been published as a full paper to date.

In summary, three out of the four studies were only carried out in pediatric populations. With regard to the influence of SNPs on PPI response, one study provided data on the induction of remission, two provided data on the maintenance of remission after a decrease in dose, and the abstract contained no information at all. Furthermore, all studies were limited by the low number of patients, the heterogeneous management of dosing, the PPI drug, and the treatment length.

## 6. Conclusions and Future Perspectives

PPIs are widely recognized as a treatment option for patients with EoE, and are the first preferred alternative in many countries and sites. Previous studies are concordant with the fact that a double dosage of PPIs is effective in achieving histological response in approximately half of patients and that, in approximately 70% of these patients, remission is maintained after a reduction in dose to the standard. However, it is not possible, in the clinical setting at present, to predict which patients will respond to this therapy. Clinical and molecular variables could help in identifying some patients with reduced chances of response, but their predictive power is far from being adequate (Figure 1). No biomarkers have been established for this purpose and, although miRNA and proteomic profiles from esophageal biopsies have shown some predictive capacity, it seems unlikely that these procedures could be applied in the clinical laboratory routine.

Pharmacogenetics represents an interesting option, since it is successful for other drugs, does not require invasive testing, and may be performed in most laboratories. Currently, the effect of *CYP2C19* variants on PPI effectiveness in other diseases (such as GERD and H. pylori eradication) is well-described and included in clinical guidelines. However, the pharmacogenetics of PPIs in EoE began only recently, with just four studies focused on this issue, analyzing *CYP2C19* and *STAT6* variants in small cohorts (Figure 1). The results are promising, but limited by the low number of patients and the heterogenous characteristics of PPI treatments.

Consequently, current evidence suggests that there may be a role for genetic variants in *CYP2C19* and *STAT6* genes in determining PPI response. Additional studies including larger cohorts with adult patients, in more controlled conditions, and an analysis of more SNPs related to EoE and PPI metabolism are needed to establish whether these findings could be applied in routine clinical practice. This could lead to the creation of a polygenic risk score that may help in predicting response to PPIs.

In addition, given that CYP2C19 is involved in the metabolism of multiple drugs, drug–drug–gene interactions may also be a relevant factor. CYP2C19 inhibitors (such as fluvoxamine, fluconazole, and fluoxetine) or inducers (like rifampin) have the potential to affect PPI metabolism [80]. Since, in individuals with CYP2C19 deficiency, a higher proportion of the PPI metabolism could occur through the CYP3A4 pathway, drugs acting as inhibitors or inducers of this enzyme should be also taken into account. Furthermore, components of foods and beverages could increase or decrease CYP2C19 activity [109,110].

Therefore, the response to treatment with PPIs most likely depends on a combination of different clinical, molecular, environmental, dietary, and genetic factors, and, to date, no integrative model has been proposed to allow for an effective and personalized approach for patients.

## Figures and Tables

**Figure 1 pharmaceutics-16-00487-f001:**
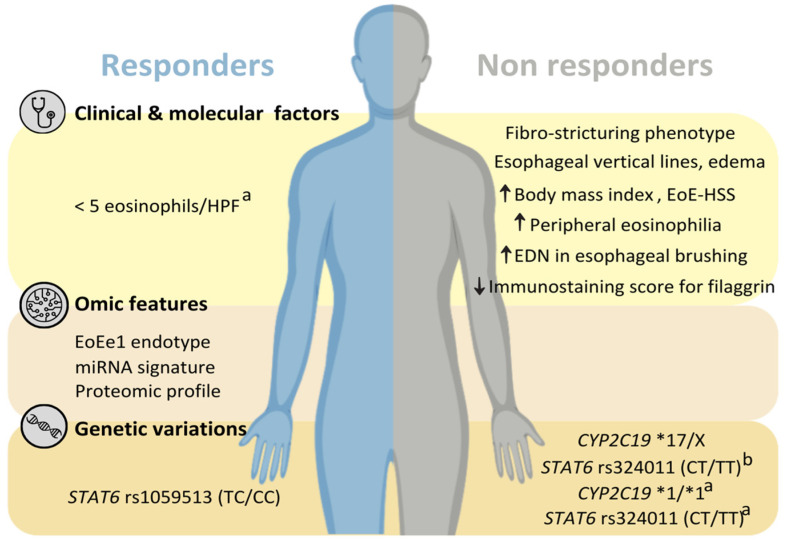
Clinical and molecular factors, omics features, and genetic variations associated by previous research with usefulness in predicting response to proton-pump inhibitors (PPI) in patients with eosinophilic esophagitis. ^a^: Only predict response after a PPI dose reduction for maintenance; ^b^: this genetic variation had a synergistic effect with the *CYP2C19*17* allele. EoE-HSS: eosinophilic esophagitis histology scoring system; HPF: high power field.

**Table 1 pharmaceutics-16-00487-t001:** Main genetic variants of the *CYP2C19* gene that affect the response to proton-pump inhibitors (PPI).

Genotype	SNP	Nucleotide Change	Amino Acid Change	Functionality
*CYP2C19*2* ^a^	rs4244285	g.24179G > A c.681G > A	Splicing defect	Non-functional protein
	rs12769205	g.17687A > G c.332-23A > G		
*CYP2C19*3*	rs4986893	g.22973G > A c.636G > A	Trp212Ter (stop gained)	Non-functional protein
*CYP2C19*4*	rs28399504	g.5026A > G c.1A > G	Met1Val (initiation codon variant)	Non-functional protein
*CYP2C19*17*	rs12248560	g.4220C > T c.-806C > T	None (upstream variant)	Increased expression of the protein

^a^ *CYP2C19*2* is represented in the table by two different SNPs, as all *CYP2C19*2* suballeles share those two SNPs. SNP: single-nucleotide polymorphism.

**Table 2 pharmaceutics-16-00487-t002:** CYP2C19 phenotypes according to the functionality of alleles.

Phenotype	Genotype	Diplotype Example
Poor metabolizers (PM)	2 non-function alleles	*CYP2C19*2/*2*
“Likely” poor metabolizers (likely PM) ^a^	1 non-function + 1 decreased-function alleles	*CYP2C19*2/*9*
Intermediate metabolizers (IM)	1 normal function + 1 non-function alleles	*CYP2C19*1/*2*
	1 non-function + 1 increased-function alleles	*CYP2C19*2/*17*
“Likely” intermediate metabolizers (likely IM) ^a^	1 normal-function + 1 decreased-function alleles	*CYP2C19*1/*9*
	1 decreased-function + 1 increased-function alleles	*CYP2C19*9/*17*
	2 decreased-function alleles	*CYP2C19*9/*9*
Normal metabolizers (NM)	2 normal-function alleles	*CYP2C19*1/*1*
Rapid metabolizers (RM)	1 normal-function + 1 increased-function alleles	*CYP2C19*1/*17*
Ultrarapid metabolizers (UM)	2 increased-function alleles	*CYP2C19*17/*17*
Indetermined metabolizers	1 or 2 uncertain-function alleles	*CYP2C19*12/*14*

^a^ There are limited data to conclusively characterize decreased-function alleles.

**Table 3 pharmaceutics-16-00487-t003:** Summary of other genetic variations, different from *CYP2C19*, that have been associated with response to proton-pump inhibitors (PPI).

Gene	SNP	Disease	Type of PPI	Functionality	Commentary
*CYP2C18*	rs2860840 rs11188059	GERD	Omeprazole	Increased (ultrarapid phenotype)	Haplotype *CYP2C:TG*
*CYP3A5*	rs776746	Healthy volunteers	Ilaprazole	Lack of activity (but increased drug clearance)	*CYP3A5*3*
*ABCB1*	rs1045642	Healthy volunteers	Lansoprazole	Decreased (lower clearance)	Lower effect than *CYP2C19* genotypes
*ATP4A*	rs2733743	Healthy volunteers	Dexlansoprazole	Increased (higher inhibition of acid gastric secretion)	Higher effect than *CYP2C19* genotypes
*STAT6*	rs167769 rs324011 rs12368672	Eosinophilic esophagitis	Esomeprazole (mainly)	Increased (higher odds of no response to PPIs during maintenance)	These three SNPs are in linkage disequilibrium; synergistic effect with *CYP2C19*17* in induction
	rs1059513			Decreased (higher odds of response to PPIs in induction)	Independent of *CYP2C19* genotypes

GERD: gastroesophageal reflux disease; SNP: single-nucleotide polymorphism.

## Data Availability

Not applicable.

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
