# Peer review of "Proton-Pump Inhibitors in Eosinophilic Esophagitis: A Review Focused on the Role of Pharmacogenetics"

_pharmaceutics, 2024, doi:10.3390/pharmaceutics16040487_

Round 1
Reviewer 1 Report
Comments and Suggestions for Authors
This is an interesting review about the role of pharmacogenetics on the eosinophilic esophagitis treatment with proton-pump inhibitors.
The manuscript is well-organized, full of details about the PPI and the EOE, including all the available data on clinical trials. However, several details need to be revised before accepting this paper for publication.
Major:
1. Explain in more detail the history of repurposing of PPI for the EOE treatment and the role of STAT6 as an alternative drug target for PPI.
2. The authors do not discuss the available GWAS —section 2.6 Prediction of PPI response by omics studies— data and the possibility of EOE subtyping according the patient´s genetic background.
3. The predictive value of a single gene genetic variation in complex phenotypes is limited, however, the creation of polygenic risk scores is best suited for this purpose, the authors must discuss this possibility for EOE and PPI. If not available, the potential strategy for implementation.
4. Please, include in the review data about potential interaction between PPI and other drugs or foods that may be used in a predictive model of response to the treatment (https://pubmed.ncbi.nlm.nih.gov/30979536/).
Minor:
1. How are the steroids for EOE treatment used (topical-line 87 or swallowed-line 109)?
2. Verify if the correct nomenclature for CYP2C19 variants are haplotypes (combination of SNPs) instead of alleles (single SNP).
3. There are at least 35 CYP2C19 haplotypes and a plethora of diplotypes with a continuous level of activity. This fact must be discussed.
4. The frequency is between 0 and 1, the frequency is not expressed as %.
5. Table 2, change dyplotype for diplotype
6. Line 292, AUC does not appear to be the correct parameter.
7. Line 339: CYP2C18: instead of CYP2:
8. Table 3. The commentaries are not clear. I do not understadnd what that means: The lower or higher effect of ABCB1 or ATP4A than the CYP2C19. Obviously STAT6 genotypes are independent from CYP2C19 genotypes are these genes are located in different genomic locations.
9. From lines 426-449. Please, include the p-values.
10. From lines 462 to 467. Exclude non-peer reviewed data (congress abstract)
Author Response
Major:
- Explain in more detail the history of repurposing of PPI for the EOE treatment and the role of STAT6 as an alternative drug target for PPI.
Thanks for these suggestion. We added new sentences to the “The evolving concept of PPI response in EoE” section of the revised manuscript to further explain the relationship between PPI response and GERD. Regarding the potential of PPI to target on STAT6 variants, 7 papers overall in literature have explored the potential effect of such drugs on JAK-STAT6 signaling pathways, exclusively related with the EoE literature. We have provided a detailed description of findings of these studies along more than 70 lines in our manuscript. Expanding the content of this section of the manuscript is a hard task for us.
- The authors do not discuss the available GWAS —section 2.6 Prediction of PPI response by omics studies— data and the possibility of EOE subtyping according the patient´s genetic background.
The first sentence in the paragraph relates to the 4 available studies using omics to differentiate responders and non-responders to PPI. We will be happy to add the information proposed by the reviewer to our manuscript, if his/her provide us with details of the studies we should review, and that can provide data that we do not know on the topic of this review.
- The predictive value of a single gene genetic variation in complex phenotypes is limited, however, the creation of polygenic risk scores is best suited for this purpose, the authors must discuss this possibility for EOE and PPI. If not available, the potential strategy for implementation.
We agree with the reviewer that the creation of a polygenic risk score could be a useful tool to predict response to PPIs in EoE once more studies have been performed about this issue. A sentence has been added at the end of the “Conclusions and future perspectives” section including this suggestion.
- Please, include in the review data about potential interaction between PPI and other drugs or foods that may be used in a predictive model of response to the treatment (https://pubmed.ncbi.nlm.nih.gov/30979536/).
The possibility that different components of the diet, coming from plant compounds, inhibit the action of CYP2C19 is really relevant. However, no available clinical practice guideline recommends combining pharmacological treatments with elimination diets, or different drugs with each other, in the treatment of patients with EoE, which downplays this issue in our review. Any case, this aspect has been commented as potentially relevant for the future in the section 6 of the revised manuscript and the suggested citation has been also included.
Minor:
- How are the steroids for EOE treatment used (topical-line 87 or swallowed-line 109)?
Topical corticosteroids for EoE are used by swallowing the compound, and then avoiding eating or drinking for a while, in order to ensure a certain contact time between the pharmaceutical product and the esophageal surface. Both terms as synonyms and used as equivalent concepts in the EoE literature. In fact, the term “swallowed topical corticosteroids” is commonly used.
- Verify if the correct nomenclature for CYP2C19 variants are haplotypes (combination of SNPs) instead of alleles (single SNP).
For some genes, as CYP2C19, the star (*) allele system is used to name the different haplotypes. These haplotypes can have a single variant or several variants that are found together. In the bibliography related to CYP2C19, the terms haplotype and allele are often used interchangeably. We have changed the term in line 232 to make clear that haplotypes and (star) alleles refer to the same in this manuscript.
- There are at least 35 CYP2C19 haplotypes and a plethora of diplotypes with a continuous level of activity. This fact must be discussed.
The sentence recommended by the reviewer has been added to the subsection 3.1.
- The frequency is between 0 and 1, the frequency is not expressed as %.
Although the reviewer is right about this, the main references used to describe the frequency of the CYP2C19 alleles/haplotypes and phenotypes in different populations report these values in % (see references #3, #77, #78 and #80). Therefore, to be accurate to the original source, we prefer to keep the values as % when describing those frequencies.
- Table 2, change dyplotype for diplotype.
This mistake has been corrected in the revised version of the manuscript.
- Line 292, AUC does not appear to be the correct parameter.
We have checked that it is the parameter mentioned in that study (see reference #75).
- Line 339: CYP2C18: instead of CYP2:
CYP2:TG is the term used in the two papers (see references #89 and #90) that analyzed the role of this haplotype in the response to PPIs so we prefer to keep it.
- Table 3. The commentaries are not clear. I do not understand what that means: The lower or higher effect of ABCB1 or ATP4A than the CYP2C19. Obviously STAT6 genotypes are independent from CYP2C19 genotypes are these genes are located in different genomic locations.
Both of them are conclusions raised by the authors of the studies that analyzed the effect of genetic variations in ABCB1 and ATP4A on PPI response. As also explained in the text, ABCB1 would have a lower effect than CYP2C19 in affecting PPI response (see reference #101) while, on the contrary, authors of the study in ATP4A gene stated that rs2733743 showed a greater effect than CYP2C19 variations on PPI response (see reference #103). Therefore, those comments in Table 3 only describe evidences found in previous studies.
- From lines 426-449. Please, include the p-values.
All p-values from Mougey et al. studies were included as suggested by the reviewer.
- From lines 462 to 467. Exclude non-peer reviewed data (congress abstract)
Since this paragraph is a summary of the results provided by the 4 studies analyzing the role of pharmacogenetics on PPI response in EoE, we found appropriate not excluding the congress abstract. We honestly think it is clearly stated which is the congress abstract, both in the paragraph describing the conclusions of the study by Bortolin et al. and also in the summary paragraph, so readers could easily identify which results came from this study that was only published as abstract.
Reviewer 2 Report
Comments and Suggestions for Authors
The manuscript is of great quality.
It describes very well the pharmacology and kinetic-metabolic aspects linked to proton pump inhibitors. Its connection with the therapeutic response in specific pathologies and the different pharmacogenetic components involved are described.
The work begins with an accurate, complete and descriptive introduction of the metabolic and pathophysiological aspects.
It addresses the link between the genotype and the activity (phenotype) of enzymes such as CYP2C19 and 3A4, their activity with the metabolism and therapeutic activity of drugs. However, other factors that modify enzymatic activity and therefore may modify the correlation between genotype and efficacy of PPIs are not fully addressed.
The source and criteria used to select the information are not adequately described. Although it is not strictly mandatory, it would greatly enrich the manuscript to provide a brief "methods" section, describing: what type of terms/search strategy was used, what reference sites were used, how citations of bibliographic origin were chosen, whether it was restricted by language and/or other filters, and whether additional selection criteria for bibliographic sources were established for this article.
More detailed information should be included on drug interactions, and actions of other xenobiotics (from the diet and/or environment) that can cause enzyme inhibition, repression, induction and/or activation phenomena. Likewise, pathological processes (infections, inflammatory diseases) and physiological processes (growth, puberty, pregnancy, hormonal changes) have an influence on the regulation of these enzymes.
The analysis of other genetic causes linked to the response to PPIs should also include an analysis of environmental and non-genetic factors.
Apart from these observations, the manuscript presents adequate academic quality, gathering and simplifying information in an orderly, clear and concise text. Use very clear data presentation resources (tables and graphs).
Author Response
Response to reviewer 2
The manuscript is of great quality.
It describes very well the pharmacology and kinetic-metabolic aspects linked to proton pump inhibitors. Its connection with the therapeutic response in specific pathologies and the different pharmacogenetic components involved are described.
The work begins with an accurate, complete and descriptive introduction of the metabolic and pathophysiological aspects.
It addresses the link between the genotype and the activity (phenotype) of enzymes such as CYP2C19 and 3A4, their activity with the metabolism and therapeutic activity of drugs. However, other factors that modify enzymatic activity and therefore may modify the correlation between genotype and efficacy of PPIs are not fully addressed.
The source and criteria used to select the information are not adequately described. Although it is not strictly mandatory, it would greatly enrich the manuscript to provide a brief "methods" section, describing: what type of terms/search strategy was used, what reference sites were used, how citations of bibliographic origin were chosen, whether it was restricted by language and/or other filters, and whether additional selection criteria for bibliographic sources were established for this article.
We immensely appreciate this reviewer's work and his/her comments on our manuscript. We are glad that the effort we put into it is recognized and as well as the potential interest for readers. This work is a literary review that does not start from a predefined PICO question, but rather addresses several aspects on the relationship between PPI treatment and EoE, trying to critically analyze the available evidence and reveal some knowledge gaps that should be answered. Some pieces of knowledge derived from diseases such as GERD have been partly extrapolated to EoE. The selection of articles did not respond to a pre-defined bibliographic search strategy (as in a systematic review), but rather each aspect of the review was based on specific literature searches, necessarily incomplete and probably biased by the expert judgment of the authors. Therefore, although adding a methodology section could improve the presentation of the manuscript, it would not modify its content in any way and would introduce information that is not necessarily true.
More detailed information should be included on drug interactions, and actions of other xenobiotics (from the diet and/or environment) that can cause enzyme inhibition, repression, induction and/or activation phenomena. Likewise, pathological processes (infections, inflammatory diseases) and physiological processes (growth, puberty, pregnancy, hormonal changes) have an influence on the regulation of these enzymes.
We appreciate this comment. All of these aspects are very relevant, but have not yet been addressed in the literature related to EoE. A new paragraph has been added to section 6 in the revised version of the manuscript to underscore the importance of addressing all these aspects in future studies.
The analysis of other genetic causes linked to the response to PPIs should also include an analysis of environmental and non-genetic factors.
Thank you for this suggestion. The amount of literature related to PPIs, its therapeutic effect, its metabolism and factors influencing it is immense. There is a reason they are one of the most prescribed pharmacological classes in the world over the last decades. Most of this information comes from acid-related diseases, and eradication treatment of H. pylori infection. However, the literature on the use of PPIs in EoE is scarce, dispersed and has less frequently addressed the interactions between its response or metabolism and the environment. We hope that new information on these aspects will allow us to update our review in the future.
Apart from these observations, the manuscript presents adequate academic quality, gathering and simplifying information in an orderly, clear and concise text. Use very clear data presentation resources (tables and graphs).
Thanks again for your insight on our manuscript.
Reviewer 3 Report
Comments and Suggestions for Authors
comprehensive review, well written no comments or concerns
Author Response
No suggestions or corrections have been raised by reviewer 3.
Thanks for reviewing our manuscript.